# Model Selection for High-Dimensional Regression under the Generalized Irrepresentability Condition

**Adel Javanmard**
Stanford University
Stanford, CA 94305
adelj@stanford.edu

**Andrea Montanari**
Stanford University
Stanford, CA 94305
montanar@stanford.edu

## Abstract

In the high-dimensional regression model a response variable is linearly related to $p$ covariates, but the sample size $n$ is smaller than $p$. We assume that only a small subset of covariates is 'active' (i.e., the corresponding coefficients are non-zero), and consider the model-selection problem of identifying the active covariates.

A popular approach is to estimate the regression coefficients through the Lasso ($\ell_1$-regularized least squares). This is known to correctly identify the active set only if the irrelevant covariates are roughly orthogonal to the relevant ones, as quantified through the so called 'irrepresentability' condition. In this paper we study the 'Gauss-Lasso' selector, a simple two-stage method that first solves the Lasso, and then performs ordinary least squares restricted to the Lasso active set.

We formulate 'generalized irrepresentability condition' (GIC), an assumption that is substantially weaker than irrepresentability. We prove that, under GIC, the Gauss-Lasso correctly recovers the active set.

## 1   Introduction

In linear regression, we wish to estimate an unknown but fixed vector of parameters $\theta_0 \in \mathbb{R}^p$ from $n$ pairs $(Y_1, X_1), (Y_2, X_2), \ldots, (Y_n, X_n)$, with vectors $X_i$ taking values in $\mathbb{R}^p$ and response variables $Y_i$ given by

$$Y_i = \langle \theta_0, X_i \rangle + W_i, \qquad W_i \sim \mathsf{N}(0, \sigma^2), \tag{1}$$

where $\langle \, \cdot \, , \, \cdot \, \rangle$ is the standard scalar product.

In matrix form, letting $Y = (Y_1, \ldots, Y_n)^\mathsf{T}$ and denoting by $\mathbf{X}$ the design matrix with rows $X_1^\mathsf{T}, \ldots, X_n^\mathsf{T}$, we have

$$Y = \mathbf{X}\,\theta_0 + W, \qquad W \sim \mathsf{N}(0, \sigma^2 \mathsf{I}_{n \times n}). \tag{2}$$

In this paper, we consider the high-dimensional setting in which the number of parameters exceeds the sample size, i.e., $p > n$, but the number of non-zero entries of $\theta_0$ is smaller than $p$. We denote by $S \equiv \mathrm{supp}(\theta_0) \subseteq [p]$ the support of $\theta_0$, and let $s_0 \equiv |S|$. We are interested in the 'model selection' problem, namely in the problem of identifying $S$ from data $Y, \mathbf{X}$.

In words, there exists a 'true' low dimensional linear model that explains the data. We want to identify the set $S$ of covariates that are 'active' within this model. This problem has motivated a large body of research, because of its relevance to several modern data analysis tasks, ranging from signal processing [9, 5] to genomics [15, 16]. A crucial step forward has been the development of model-selection techniques based on convex optimization formulations [17, 8, 6]. These formulations have lead to computationally efficient algorithms that can be applied to large scale problems. Such developments pose the following theoretical question: *For which vectors $\theta_0$, designs $\mathbf{X}$, and*

*noise levels $\sigma$, the support $S$ can be identified, with high probability, through computationally efficient procedures?* The same question can be asked for random designs $\mathbf{X}$ and, in this case, 'high probability' will refer both to the noise realization $W$, and to the design realization $\mathbf{X}$. In the rest of this introduction we shall focus –for the sake of simplicity– on the deterministic settings, and refer to Section 3 for a treatment of Gaussian random designs.

The analysis of computationally efficient methods has largely focused on $\ell_1$-regularized least squares, a.k.a. the Lasso [17]. The Lasso estimator is defined by

$$\widehat{\theta}^n(Y, \mathbf{X}; \lambda) \equiv \arg\min_{\theta \in \mathbb{R}^p} \left\{ \frac{1}{2n} \|Y - \mathbf{X}\theta\|_2^2 + \lambda \|\theta\|_1 \right\}. \tag{3}$$

In case the right hand side has more than one minimizer, one of them can be selected arbitrarily for our purposes. It is worth noting that when columns of X are in general positions (e.g. when the entries of X are drawn form a continuous probability distribution), the Lasso solution is unique [18]. We will often omit the arguments $Y, \mathbf{X}$, as they are clear from the context. (A closely related method is the so-called Dantzig selector [6]: it would be interesting to explore whether our results can be generalized to that approach.)

It was understood early on that, even in the large-sample, low-dimensional limit $n \to \infty$ at $p$ constant, $\mathrm{supp}(\widehat{\theta}^n) \neq S$ unless the columns of $\mathbf{X}$ with index in $S$ are roughly orthogonal to the ones with index outside $S$ [12]. This assumption is formalized by the so-called 'irrepresentability condition', that can be stated in terms of the empirical covariance matrix $\widehat{\Sigma} = (\mathbf{X}^\mathsf{T}\mathbf{X}/n)$. Letting $\widehat{\Sigma}_{A,B}$ be the submatrix $(\widehat{\Sigma}_{i,j})_{i \in A, j \in B}$, irrepresentability requires

$$\|\widehat{\Sigma}_{S^c, S} \widehat{\Sigma}_{S,S}^{-1} \mathrm{sign}(\theta_{0,S})\|_\infty \leq 1 - \eta, \tag{4}$$

for some $\eta > 0$ (here $\mathrm{sign}(u)_i = +1, 0, -1$ if, respectively, $u_i > 0, = 0, < 0$). In an early breakthrough, Zhao and Yu [23] proved that, if this condition holds with $\eta$ uniformly bounded away from 0, it guarantees correct model selection also in the high-dimensional regime $p \gg n$. Meinshausen and Bülmann [14] independently established the same result for random Gaussian designs, with applications to learning Gaussian graphical models. These papers applied to very sparse models, requiring in particular $s_0 = O(n^c)$, $c < 1$, and parameter vectors with large coefficients. Namely, scaling the columns of $X$ such that $\widehat{\Sigma}_{i,i} \leq 1$, for $i \in [p]$, they require $\theta_{\min} \equiv \min_{i \in S} |\theta_{0,i}| \geq c\sqrt{s_0/n}$.

Wainwright [21] strengthened considerably these results by allowing for general scalings of $s_0, p, n$ and proving that much smaller non-zero coefficients can be detected. Namely, he proved that for a broad class of empirical covariances it is only necessary that $\theta_{\min} \geq c\sigma\sqrt{(\log p)/n}$. This scaling of the minimum non-zero entry is optimal up to constants. Also, for a specific classes of random Gaussian designs (including $\mathbf{X}$ with i.i.d. standard Gaussian entries), the analysis of [21] provides tight bounds on the minimum sample size for correct model selection. Namely, there exists $c_\ell, c_u > 0$ such that the Lasso fails with high probability if $n < c_\ell \, s_0 \log p$ and succeeds with high probability if $n \geq c_u \, s_0 \log p$.

While, thanks to these recent works [23, 14, 21], we understand reasonably well model selection via the Lasso, it is fundamentally unknown what model-selection performances can be achieved with general computationally practical methods. Two aspects of of the above theory cannot be improved substantially: $(i)$ The non-zero entries must satisfy the condition $\theta_{\min} \geq c\sigma/\sqrt{n}$ to be detected with high probability. Even if $n = p$ and the measurement directions $X_i$ are orthogonal, e.g., $\mathbf{X} = \sqrt{n}\mathrm{I}_{n \times n}$, one would need $|\theta_{0,i}| \geq c\sigma/\sqrt{n}$ to distinguish the $i$-th entry from noise. For instance, in [10], the authors prove a general upper bound on the minimax power of tests for hypotheses $H_{0,i} = \{\theta_{0,i} = 0\}$. Specializing this bound to the case of standard Gaussian designs, the analysis of [10] shows formally that no test can detect $\theta_{0,i} \neq 0$, with a fixed degree of confidence, unless $|\theta_{0,i}| \geq c\sigma/\sqrt{n}$. $(ii)$ The sample size must satisfy $n \geq s_0$. Indeed, if this is not the case, for each $\theta_0$ with support of size $|S| = s_0$, there is a one parameter family $\{\theta_0(t) = \theta_0 + t\,v\}_{t \in \mathbb{R}}$ with $\mathrm{supp}(\theta_0(t)) \subseteq S$, $\mathbf{X}\theta_0(t) = \mathbf{X}\theta_0$ and, for specific values of $t$, the support of $\theta_0(t)$ is strictly contained in $S$.

On the other hand, there is no fundamental reason to assume the irrepresentability condition (4). This follows from the requirement that a specific method (the Lasso) succeeds, but is unclear why it should be necessary in general. In this paper we prove that the *Gauss-Lasso selector* has nearly optimal model selection properties under a condition that is strictly weaker than irrepresentability.

**Input:** Measurement vector $y$, design model $\mathbf{X}$, regularization parameter $\lambda$, support size $s_0$.
**Output:** Estimated support $\widehat{S}$.
 1: Let $T = \operatorname{supp}(\widehat{\theta}^n)$ be the support of Lasso estimator $\widehat{\theta}^n = \widehat{\theta}^n(y, \mathbf{X}, \lambda)$ given by

$$\widehat{\theta}^n(Y, \mathbf{X}; \lambda) \equiv \arg\min_{\theta \in \mathbb{R}^p} \left\{ \frac{1}{2n} \|Y - \mathbf{X}\theta\|_2^2 + \lambda\|\theta\|_1 \right\}.$$

 2: Construct the estimator $\widehat{\theta}^{\mathrm{GL}}$ as follows:

$$\widehat{\theta}^{\mathrm{GL}}_T = (\mathbf{X}_T^{\mathsf{T}}\mathbf{X}_T)^{-1}\mathbf{X}_T^{\mathsf{T}}y, \qquad \widehat{\theta}^{\mathrm{GL}}_{T^c} = 0.$$

 3: Find $s_0$-th largest entry (in modulus) of $\widehat{\theta}^{\mathrm{GL}}_T$, denoted by $\widehat{\theta}^{\mathrm{GL}}_{(s_0)}$, and let

$$\widehat{S} \equiv \left\{ i \in [p] : |\widehat{\theta}^{\mathrm{GL}}_i| \geq |\widehat{\theta}^{\mathrm{GL}}_{(s_0)}| \right\}.$$

We call this condition the *generalized irrepresentability condition* (GIC). The Gauss-Lasso proce-dure uses the Lasso to estimate a first model $T \subseteq \{1, \ldots, p\}$. It then constructs a new estimator by ordinary least squares regression of the data $Y$ onto the model $T$.

We prove that the estimated model is, with high probability, correct (i.e., $\widehat{S} = S$) under conditions comparable to the ones assumed in [14, 23, 21], while replacing irrepresentability by the weaker generalized irrepresentability condition. In the case of random Gaussian designs, our analysis further assumes the restricted eigenvalue property in order to establish a nearly optimal scaling of the sample size $n$ with the sparsity parameter $s_0$.

In order to build some intuition about the difference between irrepresentability and generalized irrepresentability, it is convenient to consider the Lasso cost function at 'zero noise':

$$G(\theta; \xi) \equiv \frac{1}{2n}\|\mathbf{X}(\theta - \theta_0)\|_2^2 + \xi\|\theta\|_1 = \frac{1}{2}\langle(\theta - \theta_0), \widehat{\Sigma}(\theta - \theta_0)\rangle + \xi\|\theta\|_1.$$

Let $\widehat{\theta}^{\mathrm{ZN}}(\xi)$ be the minimizer of $G(\,\cdot\,; \xi)$ and $v \equiv \lim_{\xi \to 0+} \operatorname{sign}(\widehat{\theta}^{\mathrm{ZN}}(\xi))$. The limit is well defined by Lemma 2.2 below. The KKT conditions for $\widehat{\theta}^{\mathrm{ZN}}$ imply, for $T \equiv \operatorname{supp}(v)$,

$$\|\widehat{\Sigma}_{T^c, T}\widehat{\Sigma}_{T,T}^{-1}v_T\|_\infty \leq 1.$$

Since $G(\,\cdot\,; \xi)$ has always at least one minimizer, this condition is *always satisfied*. Generalized irrepresentability requires that the above inequality holds with some small slack $\eta > 0$ bounded away from zero, i.e.,

$$\|\widehat{\Sigma}_{T^c, T}\widehat{\Sigma}_{T,T}^{-1}v_T\|_\infty \leq 1 - \eta.$$

Notice that this assumption reduces to standard irrepresentability cf. Eq. (4) if, in addition, we ask that $v = \operatorname{sign}(\theta_0)$. In other words, earlier work [14, 23, 21] required generalized irrepresentability *plus* sign-consistency in zero noise, and established sign consistency in non-zero noise. In this paper the former condition is shown to be sufficient.

From a different point of view, GIC demands that irrepresentability holds for a superset of the true support $S$. It was indeed argued in the literature that such a relaxation of irrepresentability allows to cover a significantly broader set of cases (see for instance [3, Section 7.7.6]). However, it was never clarified why such a superset irrepresentability condition should be significantly more general than simple irrepresentability. Further, no precise prescription existed for the superset of the true support.

Our contributions can therefore be summarized as follows:

- By tying it to the KKT condition for the zero-noise problem, we justify the expectation that generalized irrepresentability should hold for a broad class of design matrices.
- We thus provide a specific formulation of superset irrepresentability, prescribing both the superset $T$ and the sign vector $v_T$, that is, by itself, significantly more general than simple irrepresentability.

- We show that, under GIC, exact support recovery can be guaranteed using the Gauss-Lasso, and formulate the appropriate 'minimum coefficient' conditions that guarantee this. As a side remark, even when simple irrepresentability holds, our results strengthen somewhat the estimates of [21] (see below for details).

The paper is organized as follows. In the rest of the introduction we illustrate the range of applicability of GIC through a simple example and we discuss further related work. We finally introduce the basic notations to be used throughout the paper. Section 2 treats the case of deterministic designs $\mathbf{X}$, and develops our main results on the basis of the GIC. Section 3 extends our analysis to the case of random designs. In this case GIC is required to hold for the population covariance, and the analysis is more technical as it requires to control the randomness of the design matrix. We refer the reader to the long version of the paper [11] for the proofs of our main results and the technical steps.

## 1.1 An example

In order to illustrate the range of new cases covered by our results, it is instructive to consider a simple example. A detailed discussion of this calculation can be found in [11]. The example corresponds to a Gaussian random design, i.e., the rows $X_1^\mathsf{T}, \ldots X_n^\mathsf{T}$ are i.i.d. realizations of a $p$-variate normal distribution with mean zero. We write $X_i = (X_{i,1}, X_{i,2}, \ldots, X_{i,p})^\mathsf{T}$ for the components of $X_i$. The response variable is linearly related to the first $s_0$ covariates:

$$Y_i = \theta_{0,1} X_{i,1} + \theta_{0,2} X_{i,2} + \cdots + \theta_{0,s_0} X_{i,s_0} + W_i \,,$$

where $W_i \sim \mathsf{N}(0, \sigma^2)$ and we assume $\theta_{0,i} > 0$ for all $i \leq s_0$. In particular $S = \{1, \ldots, s_0\}$.

As for the design matrix, first $p - 1$ covariates are orthogonal at the population level, i.e., $X_{i,j} \sim \mathsf{N}(0, 1)$ are independent for $1 \leq j \leq p-1$ (and $1 \leq i \leq n$). However the $p$-th covariate is correlated to the $s_0$ relevant ones:

$$X_{i,p} = a\, X_{i,1} + a\, X_{i,2} + \cdots + a\, X_{i,s_0} + b\, \tilde{X}_{i,p} \,.$$

Here $\tilde{X}_{i,p} \sim \mathsf{N}(0, 1)$ is independent from $\{X_{i,1}, \ldots, X_{i,p-1}\}$ and represents the orthogonal component of the $p$-th covariate. We choose the coefficients $a, b \geq 0$ such that $s_0 a^2 + b^2 = 1$, whence $\mathbb{E}\{X_{i,p}^2\} = 1$ and hence the $p$-th covariate is normalized as the first $(p - 1)$ ones. In other words, the rows of $\mathbf{X}$ are i.i.d. Gaussian $X_i \sim \mathsf{N}(0, \Sigma)$ with covariance given by

$$\Sigma_{ij} = \begin{cases} 1 & \text{if } i = j, \\ a & \text{if } i = p, j \in S \text{ or } i \in S, j = p, \\ 0 & \text{otherwise.} \end{cases}$$

For $a = 0$, this is the standard i.i.d. design and irrepresentability holds. The Lasso correctly recovers the support $S$ from $n \geq c\, s_0 \log p$ samples, provided $\theta_{\min} \geq c' \sqrt{(\log p)/n}$. It follows from [21] that this remains true as long as $a \leq (1-\eta)/s_0$ for some $\eta > 0$ bounded away from 0. However, as soon as $a > 1/s_0$, the Lasso includes the $p$-th covariate in the estimated model, with high probability.

On the other hand, Gauss-Lasso is successful for a significantly larger set of values of $a$. Namely, if

$$a \in \left[0, \frac{1-\eta}{s_0}\right] \cup \left(\frac{1}{s_0}, \frac{1-\eta}{\sqrt{s_0}}\right] \,,$$

then it recovers $S$ from $n \geq c\, s_0 \log p$ samples, provided $\theta_{\min} \geq c' \sqrt{(\log p)/n}$. While the interval $((1-\eta)/s_0, 1/s_0]$ is not covered by this result, we expect this to be due to the proof technique rather than to an intrinsic limitation of the Gauss-Lasso selector.

## 1.2 Further related work

The restricted isometry property [7, 6] (or the related restricted eigenvalue [2] or compatibility conditions [19]) have been used to establish guarantees on the estimation and model selection errors of the Lasso or similar approaches. In particular, Bickel, Ritov and Tsybakov [2] show that, under such conditions, with high probability,

$$\|\widehat{\theta} - \theta_0\|_2^2 \leq C \sigma^2 \frac{s_0 \log p}{n} \,.$$

The same conditions can be used to prove model-selection guarantees. In particular, Zhou [24] studies a multi-step thresholding procedure whose first steps coincide with the Gauss-Lasso. While the main objective of this work is to prove high-dimensional $\ell_2$ consistency with a sparse estimated model, the author also proves partial model selection guarantees. Namely, the method correctly recovers a subset of large coefficients $S_L \subseteq S$, provided $|\theta_{0,i}| \geq c\sigma\sqrt{s_0(\log p)/n}$, for $i \in S_L$. This means that the coefficients that are guaranteed to be detected must be a factor $\sqrt{s_0}$ larger than what is required by our results.

An alternative approach to establishing model-selection guarantees assumes a suitable mutual incoherence conditions. Lounici [13] proves correct model selection under the assumption $\max_{i \neq j} |\widehat{\Sigma}_{ij}| = O(1/s_0)$. This assumption is however stronger than irrepresentability [19]. Candés and Plan [4] also assume mutual incoherence, albeit with a much weaker requirement, namely $\max_{i \neq j} |\widehat{\Sigma}_{ij}| = O(1/(\log p))$. Under this condition, they establish model selection guarantees for an ideal scaling of the non-zero coefficients $\theta_{\min} \geq c\sigma\sqrt{(\log p)/n}$. However, this result only holds with high probability for a 'random signal model' in which the non-zero coefficients $\theta_{0,i}$ have uniformly random signs.

The authors in [22] consider the variable selection problem, and under the same assumptions on the non-zero coefficients as in the present paper, guarantee support recovery under a cone condition. The latter condition however is stronger than the generalized irrepresentability condition. In particular, for the example in Section 1.1 it yields no improvement over the standard irrepresentability. The work [20] studies the adaptive and the thresholded Lasso estimators and proves correct model selection assuming the non-zero coefficients are of order $s_0\sqrt{(\log p)/n}$.

Finally, model selection consistency can be obtained without irrepresentability through other methods. For instance [25] develops the adaptive Lasso, using a data-dependent weighted $\ell_1$ regularization, and [1] proposes the Bolasso, a resampling-based techniques. Unfortunately, both of these approaches are only guaranteed to succeed in the low-dimensional regime of $p$ fixed, and $n \to \infty$.

## 1.3 Notations

We provide a brief summary of the notations used throughout the paper. For a matrix $A$ and set of indices $I, J$, we let $A_J$ denote the submatrix containing just the columns in $J$ and $A_{I,J}$ denote the submatrix formed by the rows in $I$ and columns in $J$. Likewise, for a vector $v$, $v_I$ is the restriction of $v$ to indices in $I$. Further, the notation $A_{I,I}^{-1}$ represents the inverse of $A_{I,I}$, i.e., $A_{I,I}^{-1} = (A_{I,I})^{-1}$. The maximum and the minimum singular values of $A$ are respectively denoted by $\sigma_{\max}(A)$ and $\sigma_{\min}(A)$. We write $\|v\|_p$ for the standard $\ell_p$ norm of a vector $v$. Specifically, $\|v\|_0$ denotes the number of nonzero entries in $v$. Also, $\|A\|_p$ refers to the induced operator norm on a matrix $A$. We use $e_i$ to refer to the $i$-th standard basis element, e.g., $e_1 = (1, 0, \ldots, 0)$. For a vector $v$, $\mathrm{supp}(v)$ represents the positions of nonzero entries of $v$. Throughout, we denote the rows of the design matrix $\mathbf{X}$ by $X_1, \ldots, X_n \in \mathbb{R}^p$ and denote its columns by $x_1, \ldots, x_p \in \mathbb{R}^n$. Further, for a vector $v$, $\mathrm{sign}(v)$ is the vector with entries $\mathrm{sign}(v)_i = +1$ if $v_i > 0$, $\mathrm{sign}(v)_i = -1$ if $v_i < 0$, and $\mathrm{sign}(v)_i = 0$ otherwise.

## 2 Deterministic designs

An outline of this section is as follows: (1) We first consider the zero-noise problem $W = 0$, and prove several useful properties of the Lasso estimator in this case. In particular, we show that there exists a threshold for the regularization parameter below which the support of the Lasso estimator remains the same and contains $\mathrm{supp}(\theta_0)$. Moreover, the Lasso estimator support is not much larger than $\mathrm{supp}(\theta_0)$. (2) We then turn to the noisy problem, and introduce the *generalized irrepresentability condition* (GIC) that is motivated by the properties of the Lasso in the zero-noise case. We prove that under GIC (and other technical conditions), with high probability, the signed support of the Lasso estimator is the same as that in the zero-noise problem. (3) We show that the Gauss-Lasso selector correctly recovers the signed support of $\theta_0$.

## 2.1 Zero-noise problem

Recall that $\widehat{\Sigma} \equiv (\mathbf{X}^\mathsf{T}\mathbf{X}/n)$ denotes the empirical covariance of the rows of the design matrix. Given $\widehat{\Sigma} \in \mathbb{R}^{p \times p}$, $\widehat{\Sigma} \succeq 0$, $\theta_0 \in \mathbb{R}^p$ and $\xi \in \mathbb{R}_+$, we define the *zero-noise Lasso estimator* as

$$\widehat{\theta}^{\mathrm{ZN}}(\xi) \equiv \arg\min_{\theta \in \mathbb{R}^p} \left\{ \frac{1}{2n} \langle (\theta - \theta_0), \widehat{\Sigma}(\theta - \theta_0) \rangle + \xi \|\theta\|_1 \right\}. \tag{5}$$

Note that $\widehat{\theta}^{\mathrm{ZN}}(\xi)$ is obtained by letting $Y = \mathbf{X}\theta_0$ in the definition of $\widehat{\theta}^n(Y, \mathbf{X}; \xi)$.

Following [2], we introduce a restricted eigenvalue constant for the empirical covariance matrix $\widehat{\Sigma}$:

$$\widehat{\kappa}(s, c_0) \equiv \min_{\substack{J \subseteq [p] \\ |J| \leq s}} \min_{\substack{u \in \mathbb{R}^p \\ \|u_{J^c}\|_1 \leq c_0 \|u_J\|_1}} \frac{\langle u, \widehat{\Sigma} u \rangle}{\|u\|_2^2}. \tag{6}$$

Our first result states that $\mathrm{supp}(\widehat{\theta}^{\mathrm{ZN}}(\xi))$ is not much larger than the support of $\theta_0$, for any $\xi > 0$.

**Lemma 2.1.** *Let $\widehat{\theta}^{\mathrm{ZN}} = \widehat{\theta}^{\mathrm{ZN}}(\xi)$ be defined as per Eq. (5), with $\xi > 0$. Then, if $s_0 = \|\theta_0\|_0$,*

$$\|\widehat{\theta}^{\mathrm{ZN}}\|_0 \leq \left( 1 + \frac{4\|\widehat{\Sigma}\|_2}{\widehat{\kappa}(s_0, 1)} \right) s_0. \tag{7}$$

**Lemma 2.2.** *Let $\widehat{\theta}^{\mathrm{ZN}} = \widehat{\theta}^{\mathrm{ZN}}(\xi)$ be defined as per Eq. (5), with $\xi > 0$. Then there exist $\xi_0 = \xi_0(\widehat{\Sigma}, S, \theta_0) > 0$, $T_0 \subseteq [p]$, $v_0 \in \{-1, 0, +1\}^p$, such that the following happens. For all $\xi \in (0, \xi_0)$, $\mathrm{sign}(\widehat{\theta}^{\mathrm{ZN}}(\xi)) = v_0$ and $\mathrm{supp}(\widehat{\theta}^{\mathrm{ZN}}(\xi)) = \mathrm{supp}(v_0) = T_0$. Further $T_0 \supseteq S$, $v_{0,S} = \mathrm{sign}(\theta_{0,S})$ and $\xi_0 = \min_{i \in S} |\theta_{0,i}/[\widehat{\Sigma}_{T_0,T_0}^{-1} v_{0,T_0}]_i|$.*

Finally we have the following standard characterization of the solution of the zero-noise problem.

**Lemma 2.3.** *Let $\widehat{\theta}^{\mathrm{ZN}} = \widehat{\theta}^{\mathrm{ZN}}(\xi)$ be defined as per Eq. (5), with $\xi > 0$. Let $T \supseteq S$ and $v \in \{+1, 0, -1\}^p$ be such that $\mathrm{supp}(v) = T$. Then $\mathrm{sign}(\widehat{\theta}^{\mathrm{ZN}}) = v$ if and only if*

$$\left\| \widehat{\Sigma}_{T^c, T} \widehat{\Sigma}_{T,T}^{-1} v_T \right\|_\infty \leq 1, \tag{8}$$

$$v_T = \mathrm{sign}\left( \theta_{0,T} - \xi \widehat{\Sigma}_{T,T}^{-1} v_T \right). \tag{9}$$

*Further, if the above holds, $\widehat{\theta}^{\mathrm{ZN}}$ is given by $\widehat{\theta}_{T^c}^{\mathrm{ZN}} = 0$ and $\widehat{\theta}_T^{\mathrm{ZN}} = \theta_{0,T} - \xi \widehat{\Sigma}_{T,T}^{-1} v_T$.*

Motivated by this result, we introduce the *generalized irrepresentability condition* (GIC) for deterministic designs.

**Generalized irrepresentability (deterministic designs).** The pair $(\widehat{\Sigma}, \theta_0)$, $\widehat{\Sigma} \in \mathbb{R}^{p \times p}$, $\theta_0 \in \mathbb{R}^p$ satisfy the generalized irrepresentability condition with parameter $\eta > 0$ if the following happens. Let $v_0$, $T_0$ be defined as per Lemma 2.2. Then

$$\left\| \widehat{\Sigma}_{T_0^c, T_0} \widehat{\Sigma}_{T_0, T_0}^{-1} v_{0, T_0} \right\|_\infty \leq 1 - \eta. \tag{10}$$

In other words we require the dual feasibility condition (8), which always holds, to hold with a positive slack $\eta$.

## 2.2 Noisy problem

Consider the noisy linear observation model as described in (2), and let $\widehat{r} \equiv (\mathbf{X}^\mathsf{T} W/n)$. We begin with a standard characterization of $\mathrm{sign}(\widehat{\theta}^n)$, the signed support of the Lasso estimator (3).

**Lemma 2.4.** *Let $\widehat{\theta}^n = \widehat{\theta}^n(y, \mathbf{X}; \lambda)$ be defined as per Eq. (3), and let $z \in \{+1, 0, -1\}^p$ with $\mathrm{supp}(z) = T$. Further assume $T \supseteq S$. Then the signed support of the Lasso estimator is given by $\mathrm{sign}(\widehat{\theta}^n) = z$ if and only if*

$$\left\| \widehat{\Sigma}_{T^c, T} \widehat{\Sigma}_{T,T}^{-1} z_T + \frac{1}{\lambda}(\widehat{r}_{T^c} - \widehat{\Sigma}_{T^c, T} \widehat{\Sigma}_{T,T}^{-1} \widehat{r}_T) \right\|_\infty \leq 1, \tag{11}$$

$$z_T = \mathrm{sign}\left( \theta_{0,T} - \widehat{\Sigma}_{T,T}^{-1}(\lambda z_T - \widehat{r}_T) \right). \tag{12}$$

**Theorem 2.5.** *Consider the deterministic design model with empirical covariance matrix $\widehat{\Sigma} \equiv (\mathbf{X}^\mathsf{T}\mathbf{X})/n$, and assume $\widehat{\Sigma}_{i,i} \leq 1$ for $i \in [p]$. Let $T_0 \subseteq [p]$, $v_0 \in \{+1, 0, -1\}^p$ be the set and vector defined in Lemma 2.2. Assume that $(i)$ $\sigma_{\min}(\widehat{\Sigma}_{T_0,T_0}) \geq C_{\min} > 0$. $(ii)$ The pair $(\widehat{\Sigma}, \theta_0)$ satisfies the generalized irrepresentability condition with parameter $\eta$. Consider the Lasso estimator $\widehat{\theta}^n = \widehat{\theta}^n(y, \mathbf{X}; \lambda)$ defined as per Eq. (3), with $\lambda = (\sigma/\eta)\sqrt{2c_1 \log p/n}$, for some constant $c_1 > 1$, and suppose that for some $c_2 > 0$:*

$$|\theta_{0,i}| \geq c_2\lambda + \lambda\big|[\widehat{\Sigma}_{T_0,T_0}^{-1}v_{0,T_0}]_i\big| \qquad \text{for all } i \in S, \tag{13}$$

$$\big|[\widehat{\Sigma}_{T_0,T_0}^{-1}v_{0,T_0}]_i\big| \geq c_2 \qquad \text{for all } i \in T_0 \setminus S. \tag{14}$$

*We further assume, without loss of generality, $\eta \leq c_2\sqrt{C_{\min}}$. Then the following holds true:*

$$\mathbb{P}\Big\{\text{sign}(\widehat{\theta}^n(\lambda)) = v_0\Big\} \geq 1 - 4p^{1-c_1}. \tag{15}$$

Note that even under standard irrepresentability, this result improves over [21, Theorem 1.(b)], in that the required lower bound for $|\theta_{0,i}|$, $i \in S$, does not depend on $\|\widehat{\Sigma}_{S,S}^{-1}\|_\infty$.

**Remark 2.6.** *Condition $(i)$ in Theorem 2.5 requires the submatrix $\widehat{\Sigma}_{T_0,T_0}$ to have minimum singular value bounded away form zero. Assuming $\widehat{\Sigma}_{S,S}$ to be non-singular is necessary for identifiability. Requiring the minimum singular value of $\widehat{\Sigma}_{T_0,T_0}$ to be bounded away from zero is not much more restrictive since $T_0$ is comparable in size with $S$, as stated in Lemma 2.1.*

We next show that the Gauss-Lasso selector correctly recovers the support of $\theta_0$.

**Theorem 2.7.** *Consider the deterministic design model with empirical covariance matrix $\widehat{\Sigma} \equiv (\mathbf{X}^\mathsf{T}\mathbf{X})/n$, and assume that $\widehat{\Sigma}_{i,i} \leq 1$ for $i \in [p]$. Under the assumptions of Theorem 2.5,*

$$\mathbb{P}\Big(\|\widehat{\theta}^{\text{GL}} - \theta_0\|_\infty \geq \mu\Big) \leq 4p^{1-c_1} + 2pe^{-nC_{\min}\mu^2/2\sigma^2}.$$

*In particular, if $\widehat{S}$ is the model selected by the Gauss-Lasso, then $\mathbb{P}(\widehat{S} = S) \geq 1 - 6\,p^{1-c_1/4}$.*

## 3 Random Gaussian designs

In the previous section, we studied the case of deterministic design models which allowed for a straightforward analysis. Here, we consider the random design model which needs a more involved analysis. Within the random Gaussian design model, the rows $X_i$ are distributed as $X_i \sim \mathsf{N}(0, \Sigma)$ for some (unknown) covariance matrix $\Sigma \succ 0$. In order to study the performance of Gauss-Lasso selector in this case, we first define the population-level estimator. Given $\Sigma \in \mathbb{R}^{p \times p}$, $\Sigma \succ 0$, $\theta_0 \in \mathbb{R}^p$ and $\xi \in \mathbb{R}_+$, the *population-level estimator* $\widehat{\theta}^\infty(\xi) = \widehat{\theta}^\infty(\xi; \theta_0, \Sigma)$ is defined as

$$\widehat{\theta}^\infty(\xi) \equiv \arg\min_{\theta \in \mathbb{R}^p} \left\{ \frac{1}{2}\langle(\theta - \theta_0), \Sigma(\theta - \theta_0)\rangle + \xi\|\theta\|_1 \right\}. \tag{16}$$

In fact, the population-level estimator is obtained by assuming that the response vector $Y$ is noiseless and $n = \infty$, hence replacing the empirical covariance $(\mathbf{X}^\mathsf{T}\mathbf{X}/n)$ with the exact covariance $\Sigma$ in the lasso optimization problem (3). Note that the population-level estimator $\widehat{\theta}^\infty$ is deterministic, albeit $\mathbf{X}$ is a random design. We show that under some conditions on the covariance $\Sigma$ and vector $\theta_0$, $T \equiv \text{supp}(\widehat{\theta}^n) = \text{supp}(\widehat{\theta}^\infty)$, i.e., the population-level estimator and the Lasso estimator share the same (signed) support. Further $T \supseteq S$. Since $\widehat{\theta}^\infty$ (and hence $T$) is deterministic, $\mathbf{X}_T$ is a Gaussian matrix with rows drawn independently from $\mathsf{N}(0, \Sigma_{T,T})$. This observation allows for a simple analysis of the Gauss-Lasso selector $\widehat{\theta}^{\text{GL}}$.

An outline of the section is as follows: (1) We begin with noting that the population-level estimator $\widehat{\theta}^\infty(\xi)$ has the similar properties to $\widehat{\theta}^{\text{ZN}}(\xi)$ stated in Section 2.1. In particular, there exists a threshold $\xi_0$, such that for all $\xi \in (0, \xi_0)$, $\text{supp}(\widehat{\theta}^\infty(\xi))$ remains the same and contains $\text{supp}(\theta_0)$. Moreover, $\text{supp}(\widehat{\theta}^\infty(\xi))$ is not much larger than $\text{supp}(\theta_0)$. (2) We show that under GIC for covariance matrix $\Sigma$ (and other sufficient conditions), with high probability, the signed support of the Lasso estimator is the same as the signed support of the population-level estimator. (3) Following the previous steps, we show that the Gauss-Lasso selector correctly recovers the signed support of $\theta_0$.

## 3.1 The $n = \infty$ problem

Comparing Eqs. (5) and (16), the estimators $\widehat{\theta}^{\mathrm{ZN}}(\xi)$ and $\widehat{\theta}^{\infty}(\xi)$ are defined in a very similar manner (the former is defined with respect to $\widehat{\Sigma}$ and the latter is defined with respect to $\Sigma$). It is easy to see that $\widehat{\theta}^{\infty}$ satisfies the properties stated in Section 2.1 once we replace $\widehat{\Sigma}$ with $\Sigma$.

## 3.2 The high-dimensional problem

We now consider the Lasso estimator (3). Recall the notations $\widehat{\Sigma} \equiv (\mathbf{X}^{\mathsf{T}}\mathbf{X})/n$ and $\widehat{r} \equiv (\mathbf{X}^{\mathsf{T}}W)/n$. Note that $\widehat{\Sigma} \in \mathbb{R}^{p \times p}$, $\widehat{r} \in \mathbb{R}^{p}$ are both random quantities in the case of random designs.

**Theorem 3.1.** *Consider the Gaussian random design model with covariance matrix $\Sigma \succ 0$, and assume that $\Sigma_{i,i} \leq 1$ for $i \in [p]$. Let $T_0 \subseteq [p]$, $v_0 \in \{+1, 0, -1\}^p$ be the deterministic set and vector defined in Lemma 2.2 (replacing $\widehat{\Sigma}$ with $\Sigma$), and $t_0 \equiv |T_0|$. Assume that $(i)$ $\sigma_{\min}(\Sigma_{T_0,T_0}) \geq C_{\min} > 0$. $(ii)$ The pair $(\Sigma, \theta_0)$ satisfies the generalized irrepresentability condition with parameter $\eta$. Consider the Lasso estimator $\widehat{\theta}^n = \widehat{\theta}^n(y, \mathbf{X}; \lambda)$ defined as per Eq. (3), with $\lambda = (4\sigma/\eta)\sqrt{c_1 \log p/n}$, for some constant $c_1 > 1$, and suppose that for some $c_2 > 0$:*

$$|\theta_{0,i}| \geq c_2\lambda + \frac{3}{2}\lambda\big|[\Sigma_{T_0,T_0}^{-1} v_{0,T_0}]_i\big| \qquad \text{for all } i \in S, \tag{17}$$

$$\big|[\Sigma_{T_0,T_0}^{-1} v_{0,T_0}]_i\big| \geq 2c_2 \qquad \text{for all } i \in T_0 \setminus S. \tag{18}$$

*We further assume, without loss of generality, $\eta \leq c_2\sqrt{C_{\min}}$. If $n \geq \max(M_1, M_2)t_0 \log p$ with $M_1 \equiv (74c_1)/(\eta^2 C_{\min})$, and $M_2 \equiv c_1(32/(c_2 C_{\min}))^2$, then the following holds true:*

$$\mathbb{P}\Big\{\mathrm{sign}(\widehat{\theta}^n(\lambda)) = v_0\Big\} \geq 1 - pe^{-\frac{n}{10}} - 6e^{-\frac{t_0}{2}} - 8p^{1-c_1}. \tag{19}$$

Note that even under standard irrepresentability, this result improves over [21, Theorem 3.(ii)], in that the required lower bound for $|\theta_{0,i}|$, $i \in S$, does not depend on $\|\Sigma_{S,S}^{-1/2}\|_\infty$.

**Remark 3.2.** *Condition $(i)$ follows readily from the restricted eigenvalue constraint, i.e., $\kappa_\infty(t_0, 0) > 0$. This is a reasonable assumption since $T_0$ is not much larger than $S_0$, as stated in Lemma 2.1 (replacing $\widehat{\Sigma}$ with $\Sigma$). Namely, $s_0 \leq t_0 \leq (1 + 4\|\Sigma\|_2/\kappa(s_0, 1))s_0$.*

Below, we show that the Gauss-Lasso selector correctly recovers the signed support of $\theta_0$.

**Theorem 3.3.** *Consider the random Gaussian design model with covariance matrix $\Sigma \succ 0$, and assume that $\Sigma_{i,i} \leq 1$ for $i \in [p]$. Under the assumptions of Theorem 3.1, and for $n \geq \max(M_1, M_2)t_0 \log p$, we have*

$$\mathbb{P}\Big(\|\widehat{\theta}^{\mathrm{GL}} - \theta_0\|_\infty \geq \mu\Big) \leq pe^{-\frac{n}{10}} + 6e^{-\frac{s_0}{2}} + 8p^{1-c_1} + 2pe^{-nC_{\min}\mu^2/2\sigma^2}.$$

*Moreover, letting $\widehat{S}$ be the model returned by the Gauss-Lasso selector, we have*

$$\mathbb{P}(\widehat{S} = S) \geq 1 - pe^{-\frac{n}{10}} - 6e^{-\frac{s_0}{2}} - 10p^{1-c_1}.$$

**Remark 3.4. [Detection level]** *Let $\theta_{\min} \equiv \min_{i \in S}|\theta_{0,i}|$ be the minimum magnitude of the non-zero entries of vector $\theta_0$. By Theorem 3.3, Gauss-Lasso selector correctly recovers $\mathrm{supp}(\theta_0)$, with probability greater than $1 - pe^{-\frac{n}{10}} - 6e^{-\frac{s_0}{2}} - 10p^{1-c_1}$, if $n \geq \max(M_1, M_2)t_0 \log p$, and*

$$\theta_{\min} \geq C\sigma\sqrt{\frac{\log p}{n}}\left(1 + \|\Sigma_{T_0,T_0}^{-1}\|_\infty\right), \tag{20}$$

*for some constant $C$. Note that Eq. (20) follows from Eqs. (17) and (18).*

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
