[Reviews · NeurIPS 2013]

Submitted by Assigned_Reviewer_2

The paper deals with model selection properties of Gauss-Lasso procedure. This is a two step procedure. In the first step a lasso estimator is fitted. In the second step, the OLS estimator is fitted on the subset of selected variables. The estimated support is given by choosing the largest s components (in absolute value) of the OLS estimator.

The main contribution of the paper is providing the generalized irrepresentability condition and showing that the Gauss-Lasso procedure correctly recovers the support of the parameter vector under this condition.

First, I believe that exposition of the material could be dramatically improved. Section 3.1 does not seem to be needed. One could formulate an optimization procedure that generalizes both (5) and (16). Then state results for this optimization procedure. The way material is currently presented, you are repeating the same things twice. More importantly, you should try to explain how do you improve the results of [1]. In particular, without going through the details of the proof, a reader should get a sense, at a higher level, of what novel tools does one need to use to improve existing results. Maybe provide an outline of the proof and point out where does your work differ from [1].

There are other two step procedures that are able to select variables consistently. See for example [2] and [3]. Both papers discuss variable selection under weaker conditions than irrepresentable condition. How does generalized irrepresentable condition compare to conditions imposed in that work.

I believe that there is another question worth answering. How does Gauss-Lasso procedure perform when the unknown parameter vector is approximately sparse? Would the procedure still select the s largest in absolute value components?

[1] M.J. Wainwright, Sharp thresholds for high-dimensional and noisy sparsity recovery using l1-constrained quadratic programming, IEEE Trans. on Inform. Theory 55 (2009)

[2] Fei Ye, Cun-Hui Zhang. Rate Minimaxity of the Lasso and Dantzig Selector for the lq Loss in lr Balls. 11(Dec):3519−3540, 2010.

[3] Sara van de Geer, Peter Bühlmann, and Shuheng Zhou. The adaptive and the thresholded Lasso for potentially misspecified models (and a lower bound for the Lasso). Electron. J. Statist. Volume 5 (2011), 688-749.
Summary: The paper studies an important problem. Exposition of the material could be dramatically improved. The authors should also compare their results to other work on two step procedures.

Submitted by Assigned_Reviewer_5

This is a theory paper on model selection in the large p small n scenario. The authors developed a generalized irrepresentability condition and applied it for studying the so-called GAUSS-LASSO selector. This paper is on the theoretical studies of the LASSO type selector, and some experimental results were provided in the appendix.

The main contribution of this paper is the proposed generalized irrepresentability condition which provides a weaker condition than the widely used irrepresentability condition. It is motivated by studying the KKT conditions of LASSO in the noiseless case. The authors applied this proposed generalized irrepresentability condition to study the GAUSS-LASSO selector. The GAUSS-LASSO selector performs a LASSO selection, then a least squares estimation on the LASSO selected variables, and sets the selected variables as the leading ones of the least squares fit. More specifically, the authors showed that there exists a threshold for the regularization parameter below which the support of the Lasso estimator remains the same and contains the support for the ground truth, and the authors established the theoretical results for LASSO in both deterministic and random designs.

Theorem 2 showed that the support of the signed support of the Lasso estimator is the same as that in the zero-noise problem with high probability. However, it is worthwhile to note that the conditions in Eq. (13-14) might not always hold. For example, when the noisy level is high, one may not find a suitable lambda or c_1 > 1 that satisfies Eq. (13).

Minor comments:
1. Lemma 2.1, Eq. (16) => Eq. (5)
2. v_0, T_0 is not easy to understand from the discussion in Section 1.3, although it should denote the restriction of v_0 to the indices in T_0.

After reading other reviewers' comments and the author response, the reviewer would like to keep the original recommendation.
Summary: This an interesting theory paper on model selection in the large p small n scenario. The authors developed a generalized irrepresentability condition and applied it for studying the so-called GAUSS-LASSO selector.

Submitted by Assigned_Reviewer_6

This paper provides a thorough and comprehensive study of
the post-Lasso (the estimator obtained by fitting a
least square model on the variables selected by the Lasso)
in the context of high dimensional sparse regression.

A new theoretical guarantee is provided for the post-
Lasso, as well as simulated experiments on a toy example
where the benefit of the procedure is clear.

Though, there are still a few points that could be improved.

First, some important references are missing about other works
considering the post-Lasso on a theoretical level
(or a variation of it):

"Pivotal Estimation of Nonparametric Functions via Square-root Lasso"
Alexandre Belloni, Victor Chernozhukov, Lie Wang
(cf. Theorem 7 for instance)


"L1-Penalized Quantile Regression in High-Dimensional Sparse Models",
Alexandre Belloni and Victor Chernozhukov, 2011
(cf. post-l1-QR Theorem 5)



Moreover, a recent work focusing on the practical properties of the
post-Lasso for a particular set of simulated data, challenges, in certain
situations, the theoretical benefit illustrated by the authors.
Can they comment on that?

"Trust, but verify: benefits and pitfalls of least-squares refitting in high dimensions",2013
Johannes Lederer




Points to correct:

-l134: without further assumptions the minimizers of G might not be unique.
the results is true under some more assumptions on the Gramm matrix,
as is well known (and in a way proved later by the authors) since

"On Sparse Representations in Arbitrary Redundant Bases", J-J. Fuchs,2004
and more recently
"The Lasso Problem and Uniqueness", Ryan J. Tibshirani, 2013

The problem occurs many time in the proof: the unicity is sometimes used before it
is proved. Adapting the results from the aforementioned papers, I encourage
the authors to show that (under suitable assumptions) unicity holds
(cf. for instance l1034, where the strict inequality
is given without any justification, see also l1071 and l1113) and therefore that
there proof is right.

I encourage the authors to fix this for the sake of clarity: it could also be better to add
an assumption mentioning when one needs the (correctly) extracted Gramm matrix to be invertible.


-l303: t_0 is defined but nowhere used in this section, and then re-used in the next one... please remove.

-l307: A comment could be added on the fact that the lambda parameter depends on an unknown quantity, eta.

-l317: What is the benefit of the assumptions w.r.t [23]? It does not seem straightforward which one is weaker:
on the one hand we need a matrix invertible of larger size (T_0 contains S) but on the other hand
only the sup-norm of a vector should be control is the proposed work.

-Section 3.1 is for me useless, the results are exactly the same as in Section 2.1.
Please remove and spare some room for the comments, and the missing references.
For instance, further comparisons on the differences between the deterministic case
and the random case could be investigated.

-l408: Did the authors try to improve the term depending on exp(-t_0/2)?
It seems to be the weak part of the probability control provided in Theorem 3.4

-l441/443: Candes and Cand\'es are mispelled. It should be Cand\`es

-l553: it seems there is a sign issue.


General questions:

- can the authors comment on the fact that the sparsity level must be known a prior (no adaptivity) in their procedure?
-When is T_0=S (l267)? That would be interesting to understand when the two are identical.


Summary: Overall, the paper is clear, sharp and is of high interest
for statisticians and practitioners.
Author Feedback

Author rebuttal: We thank the reviewers for their detailed and thoughtful comments. Responses to the comments follow.

*REVIEWER 2:

We will try our best to improve the exposition of the martial. Apart from the paper organization, the only concern raised is the relation of our work with [1] (Wainwright, 2009), [2], (Ye, Zhang, 2010) and [3] (van de Geer et al. 2011).

Regarding [1]: We are analyzing a different model selection method, and prove that it is successful under *much weaker* conditions than the Lasso, analyzed in [1].

Regarding [2]: This paper requires the `cone invertibility factor' to be of order 1
to guarantee support recovery under the same conditions on the non-zero coefficients as in our paper. This assumption is stronger than generalized irrepresentability. In particular, for the simple example in Section 1.1 (one confounding factor), it yields no improvement over standard irrepresentability (it still needs correlation coefficient $a$ to be smaller than $(1-\eta)/s0$ for some $\eta > 0$). By contrast, our results yield substantial improvement in that $a$ can be as large as $(1-\eta)/sqrt(s0)$.

Regarding [3]: The results that compare most directly with our work are Lemma 3.3 and Corollary 3.2. These require the non-zero coefficients to be of order s_0\sqrt{(\log p)/n} (see also the discussion following corollary 3.2 therein). This is a very strong condition. As a comparison, our work recovers the support under the ideal scaling \sqrt{\log p/n}.

- Regarding your question about approximately sparse vectors, one can decompose theta0 into theta1 (the s largest entries) and theta2 (the remaining ones). Then, y = X*theta1+X*theta2+w. If the gap between the entries in theta1 and theta2 is large enough, then Gauss-Lasso treats the term (X*theta2+w) as noise and recovers theta1.


*REVIEWER 5:

We agree with the referee about the conditions in Eq. (13-14). For instance Eq. (13) imposes a condition on the nonzero entries of theta0. Namely, the coefficients have to be larger than \lambda that is approximately \sigma\sqrt{(\log p)/n}. Note that this is the order optimal detection level (i.e. the same as for orthogonal designs).

*REVIEWER 6:

For the sake of space (and since the report is very positive), we limit ourselves to answering a subset of the questions:
- We agree that a comparison with work by Belloni et al. and by Lederer (published at time of submission) would be useful.

- Unicity holds when columns of X are in general positions (e.g. when the entries of X are drawn form continuous probability distributions). We will point this out in the revision.

- The main step forward with respect to [23] is the use of generalized irrepresentability instead of simple irrepresentability. As for condition (13), under irrepresentability T_0 = S, and hence this condition is strictly weaker than the analogous condition in [23]. If --for instance-- the signs of \theta_0 are random
(as in Candes, Plan, 2009), then this condition becomes much weaker than the one in [23].

- The method can be adapted to cases where s0 is unknown by modifying step (3), and thresholding at a level that depends on the noise level. More specifically, choosing mu = O(sigma \sqrt((log p)/n)) in theorem 2.7, if theta_min > 2*mu, we can obtain the nonzero entries (w.h.p) by thresholding at mu.